# Phenethyl Isothiocyanate Protects against High Fat/Cholesterol Diet-Induced Obesity and Atherosclerosis in C57BL/6 Mice

**DOI:** 10.3390/nu12123657

**Published:** 2020-11-27

**Authors:** Min-Hee Gwon, Young-Sun Im, A-Reum Seo, Kyoung Yun Kim, Ha-Rin Moon, Jung-Mi Yun

**Affiliations:** 1Nutrition Education Major, Graduate School of Education, Chonnam National University, Gwangju 61186, Korea; a961217@naver.com; 2Department of Food and Nutrition, Chonnam National University, Gwangju 61186, Korea; zz6z6@naver.com (Y.-S.I.); 01031855015@hanmail.net (A.-R.S.); kimkyjs0906@gmail.com (K.Y.K.); cjsghk1970@naver.com (H.-R.M.)

**Keywords:** phenethyl isothiocyanate, atherosclerosis, obesity, histone modifications, lipid accumulation, inflammation, reverse cholesterol transport

## Abstract

This study concerns obesity-related atherosclerosis, hyperlipidemia, and chronic inflammation. We studied the anti-obesity and anti-atherosclerosis effects of phenethyl isothiocyanate (PEITC) and explored their underlying mechanisms. We established an animal model of high fat/cholesterol-induced obesity in C57BL/6 mice fed for 13 weeks. We divided the mice into five groups: control (CON), high fat/cholesterol (HFCD), HFCD with 3 mg/kg/day gallic acid (HFCD + G), and HFCD with PEITC (30 and 75 mg/kg/day; HFCD + P30 and P75). The body weight, total cholesterol, and triglyceride were significantly lower in the HFCD + P75 group than in the HFCD group. Hepatic lipid accumulation and atherosclerotic plaque formation in the aorta were significantly lower in both HFCD + PEITC groups than in the HFCD group, as revealed by hematoxylin and eosin (H&E) staining. To elucidate the mechanism, we identified the expression of genes related to inflammation, reverse cholesterol transport, and lipid accumulation pathway in the liver. The expression levels of peroxisome proliferator activated receptor gamma (PPARγ), liver-X-receptor α (LXR-α), and ATP binding cassette subfamily A member 1 (ABCA1) were increased, while those of scavenger receptor A (SR-A1), cluster of differentiation 36 (CD36), and nuclear factor-kappa B (NF-κB) were decreased in the HFCD + P75 group compared with those in the HFCD group. Moreover, PEITC modulated H3K9 and H3K27 acetylation, H3K4 dimethylation, and H3K27 di-/trimethylation in the HFCD + P75 group. We, therefore, suggest that supplementation with PEITC may be a potential candidate for the treatment and prevention of atherosclerosis and obesity.

## 1. Introduction

Obesity is associated with atherosclerosis, which is characterized by the infiltration of lipid particles into the arterial wall [1]. The liver is a central organ of lipid metabolism and plasma lipoprotein synthesis [2]. Fatty liver shares many risk factors with cardiovascular disease (CVD), with a close relationship between fatty liver and adverse cardiovascular events, such as atherosclerosis and hypertension [3].

Hepatic steatosis is extremely prevalent in obese individuals, and lipid accumulation in the liver is generally highly correlated with the risk of CVD [4]. Although several proteins may contribute to this process, scavenger receptor A (SR-A) and cluster of differentiation 36 (CD36) have been demonstrated to be most relevant for uptake of fatty acids [5]. Hepatic CD36 expression is significantly elevated and is considered to be positively correlated with liver fat content in vivo [6].

Atherosclerosis leads to endothelial cell dysfunction, which causes a chronic inflammatory reaction accompanied by the recruitment of immune cells [7]. Lipid metabolism disorders, particularly reverse cholesterol transport (RCT)-related disorders, are critical steps in the pathogenesis and development of atherosclerosis [8]. This pathway is a process that occurs when high-density lipoprotein (HDL) transports cholesterol and cholesterol esters from peripheral tissues and cells to the liver, with excretion as bile acids [8]. It is deeply related to the activation of the ATP binding cassette subfamily A member 1 (ABCA1), liver-X-receptor α (LXR-α), and peroxisome proliferator activated receptor gamma (PPARγ). A recent study reported these genes in lipid regulation, inflammation, and apoptosis, which seems to be the center of atherosclerotic plaque progression in vitro and in vivo [9]. Thus, scavenger receptors and PPAR-LXR-ABCA1 interactions are essential for inflammation, atherosclerosis, and cholesterol homeostasis [9].

In addition, nuclear factor-kappa B (NF-κB) transcription factor has been linked to diverse inflammatory diseases such as atherosclerosis. Its activation-mediated signaling has been detected at different stages of atherosclerosis, beginning with plaques formation, destabilization, and rupture [10].

The epigenetic mechanisms include DNA methylation, histone modifications, and micro-RNA processes. Disruption of this balance may lead to several pathologies and contribute to obesity [11]. Recent studies demonstrated that hyperglycemia increases H3K9ac and H3K4me2 and decreases H3K9me2, thereby enhancing monocyte activation, allowing NF-κB to bind to inflammatory genes [12]. Epigenetic modulators of dietary components, such as fisetin [13], sulforaphane [14], curcumin [15], and epigallocatechin gallate [16], suppress proinflammatory responses through transcription factor activation. The epigenetic mechanism of atherosclerosis is being intensively studied and research to understand the pathogenesis of atherosclerosis is ongoing. Nevertheless, the analysis of specific cell types in the complex environment of plaques remains difficult [17].

Recent studies have reported that many dietary supplements or dietary products are likely to be developed as therapeutics. These include red wine, propolis, apigenin, and berries, which attenuate the progression of atherosclerosis [18]. In addition, dietary intake of walnuts, which have antioxidant activity, is beneficial for lipid metabolism and atherosclerotic plaque development, as evidenced by the decreased CD36 expression in apoE-/- mice fed a high fat diet (HFD) [19]. Recent evidence suggested that dietary quercetin may be effective in lowering the risk of atherosclerosis by upregulating PPARγ, LXR-α, and ABCA1 and downregulating CD36 in the liver of apoE-/- mice fed HFD [8].

Cruciferous vegetables, including broccoli, watercress, cauliflower, and cabbage, play a protective role in a variety of environments because of organic isothiocyanates (ITCs). ITC-containing compounds are characterized by sulfur-containing N=C=S functional groups. ITCs have been studied extensively in the prevention and treatment of metabolic diseases. Phenethyl isothiocyanate (PEITC) is one of the most extensively studied ITCs. PEITC is a major constituent of watercress and other cruciferous vegetables. PEITC has anti-cancer mechanisms that include the induction of apoptosis and/or inhibition of proliferation [20,21,22,23]. However, the potential use of PEITC to alleviate obesity and atherosclerosis in mice fed a high fat/cholesterol diet (HFCD) as well as its mechanism of action and histone modification activity remain unexplored.

In this study, we aimed to investigate the anti-obesity and anti-atherosclerosis effects of PEITC supplementation in C57BL/6 mice fed HFCD. Previous studies have reported that gallic acid (GA, 3,4,5-trihydroxybenzoic acid) reduces body, liver, and adipose tissue weights in a rat model of diet-induced obesity [24,25]. GA has potential protective effects that include anti-cancer, anti-inflammatory, and antioxidant activities [26]. Several reports have described that GA inhibits NF-kB activation in vivo [27]. In addition, and in agreement with our findings, mice fed GA displayed decreased fat droplets throughout the liver [28]. Moreover, GA is an individually recognized health functional food by the Korean government based on its ability to improve cholesterol levels. Therefore, we used GA as a positive control. In this study, we hypothesized that HFCD supplementation with PEITC in mice could reduce fat/cholesterol in the liver and aorta and has a protective effect against obesity and atherosclerosis. To test our hypothesis, we used C57BL/6 mice with high fat/cholesterol-induced obesity and analyzed the effect of PEITC supplementation through the expression of genes related to inflammation, reverse cholesterol transport, and lipid accumulation pathways in the liver. Furthermore, we analyzed H3K9 and H3K27 acetylation, H3K4 dimethylation, and H3K27 di-/trimethylation with modulated PEITC supplementation.

## 2. Materials and Methods

### 2.1. Animals and Diet

Male C57BL/6 mice (4 weeks old) were purchased from Orient Bio Co. (Gyeonggi-do, South Korea). The animals were housed under controlled laboratory conditions (20–25 °C; humidity, 55–60%), under a 12 h shift of the light/dark cycle, with free access to food and water. The animals were maintained on the AIN-93G diet (Saeronbio Inc., Gyeonggi-do, South Korea) for 1 week prior to initiation of the diet studies. Then, the animals were randomly divided into five groups and fed for 13 weeks with the following (*n* = 10): normal diet with AIN-93G diet (control, CON), high fat diet (60% calories from fat) containing 1% cholesterol (HFCD), HFCD with 3 mg/kg/day GA (HFCD + G), HFCD with 30 mg/kg/day PEITC (HFCD + P30), or HFCD with 75 mg/kg/day PEITC (HFCD + P75). GA and PEITC were procured from Sigma-Aldrich (St. Louis, MO, USA). Body weight and food intake were assessed once every 3 days. Experiments were approved by the Animal Care and Use Committee of Chonnam National University (CNU IACUC-YB-2019-01).

### 2.2. Collection of Serum and Tissue Samples

Prior to sacrifice, the total body weight, food intake, and fasting blood glucose levels were measured. After fasting overnight for 12 h, each mouse was deeply anesthetized with approximately 5 mL of diethyl ether. Blood samples were collected from the postcaval vein of each animal immediately after sacrifice and centrifuged at 870× *g* for 5 min at 4 °C. After sacrifice, each tissue was immediately excised and weighed. Serum samples and tissues (liver, epididymal fat, retroperitoneal adipose, and kidney tissues) were stored at −80 °C before analysis.

### 2.3. Biochemical Analyses

After collecting blood, complete gross postmortem examinations were performed. Aspartate aminotransferase (AST), alanine aminotransferase (ALT), total cholesterol (TC), triglycerides (TG), low-density lipoprotein cholesterol (LDL-C), and high-density lipoprotein cholesterol (HDL-C) levels were determined using commercial kits (Asan Pharmaceutical, Seoul, South Korea). Non-esterified fatty acid (NEFA) was measured using assay kits (Wako Pure Chemicals, Tokyo, Japan). Atherogenic index (AI) values were calculated using the standard formula: AI = log_10_ (TG/HDL-C) [29].

### 2.4. Analyses of Blood Glucose and Insulin

Insulin enzyme-linked immunosorbent assay kit (Morinaga Institute of Biological Science, Yokohama, Japan) was used to measure serum insulin. The portable blood glucose meter CareSens^®^ N (i-SENS, Inc., Seoul, Korea) was used to measure fasting blood glucose, according to the method described by Shi et al. [30]. Homeostasis model assessment for insulin resistance (HOMA-IR) values were calculated using the following formula: HOMA-IR = fasting insulin (μU/mL) × fasting glucose (mmol/L)/22.5.

### 2.5. Histopathological Examination of Aorta and Liver Tissues

The aorta and liver tissues from representative mice of each group were preserved in 10% buffered formalin solution at pH 7.0, washed with phosphate buffered saline, and embedded in paraffin. Paraffin blocks were sectioned to a thickness of approximately 4–5 µm, mounted on slides, and stained with hematoxylin and eosin (H&E). The sections were observed under a light microscope at 200× *g* or 400× *g* magnification.

### 2.6. Preparation of Whole Cell and Nuclear Lysates

Whole cell lysates were prepared using RIPA buffer (Biosesang, Gyeonggi-do, South Korea) and Halt™ protease and phosphatase inhibitor cocktail (Thermo Fisher Scientific, Waltham, MA, USA). Nuclear lysates were prepared using nuclear extraction buffer (20 mM HEPES, 0.4 mM NaCl, 1 mM ethylenediaminetetraacetic acid (EDTA), 1 mM EGTA, 1 mM dithiothreitol, 1 mM phenylmethylsulfonyl fluoride (PMSF), and 10% NP-40). The lysates were collected and cleared by centrifugation at 16,343× *g* for 1 min at 4 °C, and the supernatant was aliquoted and stored at −80 °C. The protein concentration of the lysates was measured by the bicinchoninic acid protein assay (Thermo Fisher Scientific).

### 2.7. Histone Extraction and Purification

Core histones were purified from liver tissue using a Histone Purification Mini Kit (40026; Active Motif, Carlsbad, CA, USA) according to the manufacturer’s protocol. Tissue samples were homogenized in ice-cold extraction buffer using a dounce homogenizer and a rotating platform. The extract was centrifuged at 16,000× *g* for 5 min at 4 °C and the pH was adjusted to 8 using a neutralization buffer. After equilibrating the spin column with equilibration buffer, the remaining crude histone extract was passed through the column. The extract was washed three times with wash buffer and eluted with elution buffer. The histones were precipitated overnight with 4% perchloric acid (PCA). The next day, the samples were centrifuged at 16,000× *g* for 1 h at 4 °C and washed with acetone and PCA. The purified histones were resuspended in diethyl pyrocarbonate water, quantified, and used for analysis.

### 2.8. Immunoblotting

For immunoblotting, 10–40 μg protein was separated on 8–15% polyacrylamide gels and transferred to nitrocellulose membrane (Merck Millipore Ltd., Tullagreen, Carrigtwohill, County Cork, Ireland). The membranes were incubated in blocking buffer containing 5% nonfat milk and then incubated with the appropriate primary antibody for 2 h at room temperature. ABCA1 (ab18180), SR-A1 (ab151707), and CD36 (ab133625) antibodies were purchased from Abcam (Cambridge, MA, USA). NF-κB (sc-8008), LXR-α (sc-377260), and β-actin (sc-47778) antibodies were purchased from Santa Cruz Biotechnology (Dallas, TX, USA). Histone H3 (4499S) and HDAC3 (3949S) antibodies were purchased from Cell Signaling Inc. (Beverly, MA, USA). PPARγ (CSB-PA003839) antibody was purchased from CUSABIO (Houston, TX, USA). After incubation with the specific primary antibody, the membranes were washed twice and incubated with the appropriate secondary antibodies for 2 h at room temperature. The membranes were washed three times and detected with Western Blotting Luminol Reagent (Santa Cruz Biotechnology) and imaged on a ChemiDoc XRS+ System (Bio-Rad). The blots were stripped and then reprobed with β-actin or Histone H3 as a loading control.

### 2.9. Quantitative Polymerase Chain Reaction (qPCR)

Total RNA was isolated from mouse livers using TRIzol reagent (Ambion, Life Technologies, Carlsbad, CA, USA) and transcribed into cDNA using Omniscript RT kit (QIAGEN, Hilden, Germany). SYBR green-based quantitative PCR was performed in 96-well plates using iQ SYBR Green Supermix (Bio-Rad) with the CFX96 Touch Real-Time PCR Detection System. The primers were procured from Bioneer (Daejeon, Korea) and primers were as follows (Forward: Fw; Reverse: Rv): mouse ABCA1: Fw 5′–TGGGAACTCCTGCTAAAAT–3′ and Rv 5′–CCATGTGGTGTGTAGACA–3′; mouse PPARγ: Fw 5′–ATTCTGGCCCACCAACTTCGG–3′ and Rv 5′–TGGAAGCCTGATGCTTTATCCCCA–3′; mouse LXR-α: Fw 5′–AGGGATAGGGTTGGAGTC–3′ and Rv 5′–TTCAAGCGGATCTGTTCT–3′; mouse NF-κB: Fw 5′–ACCACTGCTCAGGTCCACTGTC–3′ and Rv 5′–GCTGTCACTATCCCGGAGTTCA–3′; mouse β-actin: Fw 5′–ACTGCCGCATCCTCTTCCTC–3′ and Rv 5′–CTCCTGCTTGCTGATCCACATC–3′. All reactions were run in triplicate, and significance was determined by comparison with β-actin normalized 2^−ΔΔCT^ values.

### 2.10. Statistical Analysis

All experiments were performed at least three times. Data are expressed as the mean ± standard deviation (SD). The significance of differences between the groups was determined by applying one-way analysis of variance (SPSS version 25.0 software, SPSS Institute, Chicago, IL, USA), and Duncan’s multiple range test was used to compare significant differences between groups. Statistical significance was expressed at *p*-value < 0.05.

## 3. Results

### 3.1. PEITC Supplementation Significantly Attenuated Body Weight Gain and Food Efficiency in C57BL/6 Mice Fed HFCD

The weight changes in mice during the entire experiment are shown in Table 1. There was no significant intragroup variation (CON: 19.85 ± 0.33 g, HFCD: 19.87 ± 0.31 g, HFCD + G: 19.88 ± 0.3 g HFCD + P30: 19.95 ± 0.33 g, HFCD + P75: 19.87 ± 0.31 g) in the basal body weights. After 13 weeks of each diet, the HFCD group had gained significantly more weight than the CON group (*p* < 0.05). The HFCD group had gained 27.85 g and the HFCD + P75 group had gained 17.2 g compared with the baseline body weight. Mice in both HFCD + PEITC groups displayed increased food intake relative to the HFCD only group. Despite the higher intake in the HFCD + P75 group than in the HFCD group, the body weight change was significantly less (*p* < 0.05).

### 3.2. Tissue Weights in C57BL/6 Mice Fed HFCD

As shown in Figure 1 and Table 2, the HFCD group displayed significantly higher liver weights compared with the CON group (3.09 ± 0.21 vs. 1.45 ± 0.08 g; *p* < 0.05). The increased liver weight in the HFCD group was lowered significantly by PEITC supplementation (HFCD + P75 group, 2.1 ± 0.26 g; *p* < 0.05). In addition, there was a difference in adipose tissue weight between the CON group and the HFCD group; and increased retroperitoneal adipose weight was significantly less in the HFCD + 75 group than in the HFCD group (0.62 ± 0.07 vs. 1.04 ± 0.1 g; *p* < 0.05).

### 3.3. PEITC Supplementation Significantly Ameliorated HFCD-Induced Aorta and Hepatic Injury in C57BL/6 Mice

H&E staining of liver sections confirmed the protective effect of PEITC on HFCD-induced hepatic injury. Cytoplasmic vacuolation with deposition of lipid droplets was observed in the hepatocytes of the mice fed HFCD (Figure 2). These findings indicated severe hepatocyte necrosis. PEITC supplementation ameliorated the histopathological changes in the liver. The amelioration was most pronounced in mice from the HFCD + P75 group. In atherosclerosis, lipids and inflammatory cells accumulate in the tunica intima and tunica media layers, leading to their thickening and narrowing of the lumen of blood vessels [31]. After 13 weeks of HFCD supplementation, the histological evaluation revealed narrowing of the blood vessel lumen in the HFCD group compared with the CON group. The HFCD + P75 and the CON groups were very similar.

### 3.4. PEITC Supplementation Improved the Serum Lipid Profile in Mice

We measured serum ALT and AST levels to assess liver function (Table 3). Mice in the both HFCD + PEITC groups displayed lower serum levels of ALT and AST compared with the HFCD group. Although the levels were improved with PEITC supplementation, the difference was not significant. It showed that PEITC ameliorated liver injury in HFCD-induced obesity mice. Furthermore, hepatic lipid profiling showed that the TC, TG, LDL-C, and NEFA levels in the livers of the HFCD + P75 mice were lower compared with those in the livers of mice from the HFCD group. The serum TC levels of the HFCD group were significantly higher than the levels in mice from the CON group (113.13 ± 4.52 mg/dL vs. 69.18 ± 1.9 mg/dL; *p* < 0.05). The TC level was significantly lower in the HFCD + P75 group (67.06 ± 5.75 mg/dL; *p* < 0.05) compared with the HFCD group. The serum TG levels in the HFCD group were significantly increased compared with those in the CON group (94.31 ± 7.05 vs. 67.39 ± 6.94 mg/dL; *p* < 0.05). Compared with the HFCD group, the TG levels were significantly lower in the HFCD + P75 group (68.5 ± 8.45 mg/dL; *p* < 0.05). The serum LDL-C levels in the HFCD group were significantly higher than those in the CON group (46.97 ± 6.69 vs. 23.53 ± 4.9 mg/dL; *p* < 0.05). Compared with the HFCD group, the LDL-C levels were lower in both HFCD + PEITC groups. The serum HDL-C and NEFA levels were not significantly different between all groups. The AI is believed to be an important risk factor for the diagnosis of atherosclerosis. Here, the AI was increased significantly in the HFCD group compared with that in the HFCD + P75 group (0.33 ± 0.03 vs. 0.14 ± 0.04; *p* < 0.05). These results clearly indicated that PEITC can potentially reduce this risk factor.

### 3.5. PEITC Supplementation Improved the Levels of Fasting Glucose and Insulin in Serum, and the HOMA Values

We measured serum insulin, fasting glucose, and HOMA-IR levels (Table 4). The serum insulin levels in the HFCD group were significantly increased compared with those in the CON group (1.45 ± 0.15 vs. 0.47 ± 0.11 ng/mL; *p* < 0.05). Compared with those in the HFCD group, the serum insulin levels were significantly lower in the HFCD + P75 group (0.31 ± 0.19 ng/mL; *p* < 0.05). The fasting glucose levels in the HFCD group were significantly increased compared with the CON group (216 ± 15.04 vs. 146.67 ± 19.22 mg/dL; *p* < 0.05). Compared with the HFCD group, the fasting glucose levels were decreased in the HFCD + P75 group (147.67 ± 33.21 mg/dL; *p* < 0.05). The HOMA-IR value, a surrogate parameter for insulin resistance, was significantly lower in the HFCD + P75 group than in the HFCD group (0.14 ± 0.1 vs. 0.77 ± 0.06; *p* < 0.05), indicating that PEITC may contribute to the reduction of the HOMA-IR value in obese mice, and thereby improve insulin resistance.

### 3.6. PEITC Modulated the Levels of Proteins of the Reverse Cholesterol Transport Pathway in the Liver

The expression of PPARγ, LXR-α, and ABCA1 in mouse liver tissue was measured using immunoblotting and qPCR. LXR-α and ABCA1 expression was decreased in the HFCD group compared with that in the CON group (Figure 3). These levels were increased in the HFCD + P75 group. In addition, the transcription of LXR-α and ABCA1 mRNAs decreased in the HFCD group and markedly increased in the HFCD + P75 group compared with that in the HFCD group, as determined using qPCR (*p* < 0.05). Immunoblotting revealed significantly upregulated protein expression of PPARγ in the HFCD + P75 group compared with the HFCD group (*p* < 0.05). The transcription of PPARγ mRNA increased in the HFCD group compared with the CON group, but there was no significant difference. The findings indicated that PEITC modulates the levels of proteins of the reverse cholesterol transport pathway in the liver via the PPARγ-LXR-α-ABCA1 pathway.

### 3.7. PEITC Modulated the Protein Levels of Lipid Accumulation Genes in the Liver

Next, we observed the lipid accumulation effect of PEITC using immunoblotting. Expression levels of the lipid uptake-related proteins SR-A1 and CD36 were measured in mouse liver tissue. The HFCD group displayed significantly increased expression of SR-A1 and CD36 compared with the CON group (Figure 4). The HFCD + P75 group displayed significantly reduced SR-A1 and CD36 expression compared with the HFCD group. The findings indicated the importance of PEITC in suppressing the development of atherosclerosis by reducing lipid accumulation, at least by reducing the expression of scavenger receptors.

### 3.8. PEITC Modulated the Protein Levels of Inflammation Genes in the Liver

We measured the expression of the NF-κB proinflammatory transcription factor in mouse liver tissue. NF-κB expression was markedly increased in the HFCD group compared with the CON group (Figure 5). NF-κB expression was decreased in the HFCD + P75 group compared with the level in the HFCD group. This result indicated that PEITC reduces obesity-induced inflammation. In addition, qPCR revealed a significant decrease in the transcription of NF-κB mRNA in the HFCD + P75 group compared with the HFCD group (*p* < 0.05). The findings indicated that PEITC inhibits inflammation by modulating NF-κB.

### 3.9. PEITC Supplementation Abrogated the HFCD-Induced Increase in Total H3K4 Dimethylation, H3K27 Di-/Trimethylation, and H3K9 and H3K27 Acetylation Expression

We extracted histones from mouse liver tissues and investigated the status of histone modifications by immunoblotting using histone antibodies (Figure 6). We evaluated histone H3K9 and H3K27 acetylation, H3K4 dimethylation, and H3K27 di-/trimethylation. Total H3K4 dimethylation and H3K27 di-/trimethylation were increased in the livers of mice fed HFCD. The increases in total H3K4 dimethylation and H3K27 di-/trimethylation of liver tissue were reduced with PEITC supplementation (HFCD + P75). In addition, H3K9 and H3K27 acetylation increased in the HFCD group, whereas PEITC supplementation decreased H3K9 acetylation and H3K27 acetylation.

### 3.10. PEITC Supplementation Abrogated the HFCD-Induced Increase in HDAC3 Expression

Finally, we measured the expression level of HDAC3 in mouse liver tissue. The expression of HDAC3 was markedly increased in the HFCD group compared with the CON group (Figure 7). However, HDAC3 expression was markedly decreased in the HFCD + P75 group compared with that in the HFCD group. HDAC3 is essential for the induction of a proinflammatory gene in response to a variety of stimuli [32]. The present and former findings indicated that PEITC reduces obesity-induced inflammation by inhibiting HDAC3.

## 4. Discussion

Atherosclerosis is a leading cause of vascular disease worldwide [33]. An important risk factor for atherosclerosis is elevated blood lipid levels [28]. Because the liver plays an important role in regulating the homeostasis of lipid metabolism and systemic inflammation, atherosclerosis has been strongly correlated with liver steatosis [34]. A strong association between dietary fatty acid intake and CVD risk factors and clinical events has been described [35]. In addition, numerous studies have indicated that obesity can be prevented by inhibiting hepatic lipogenesis [36,37]. Thus, manipulation of lipid metabolism and inflammation is a generally pursued strategy to prevent atherosclerosis associated with impaired metabolic regulation [38]. Furthermore, the effects of various genes that regulate histone modifications on the treatment of atherosclerosis have been studied recently [39].

Recently, the focus has been on the discovery of naturally occurring antioxidants of plant origin to manage disease and lower therapeutic drug-related side effects. PEITC is a constituent of many cruciferous vegetables. It has beneficial and varied physiological activities, including anti-cancer effects in glioblastoma, prostate cancer, breast cancer, and leukemia [40]. Recently PEITC reported improved HFD-induced obesity and fatty liver by downregulating adipocyte differentiation and expression of lipogenic transcription factors and enzymes [41].

The aim of this study was to assess the protective effect of PEITC against obesity and atherosclerosis in C57BL/6 mice fed HFCD. Over 13 weeks, the HFCD diet significantly increased body weight, fat mass, and serum lipid levels. These changes were appreciably alleviated in mice consuming the PEITC supplemented diet. H&E staining revealed that PEITC supplementation reduced the number of lipid droplets in liver tissue sections and alleviated atherosclerotic plaque formation in aorta tissue sections.

The C57BL/6 strain of mice is the most sensitive to atherosclerosis among all mouse strains [42]. C57BL/6 mice are susceptible to obesity when fed HFD [42]. Therefore, C57BL/6 mice are the most widely used in obesity and atherosclerosis studies. Consistent with these observations, we found that the weight gain in the HFCD group of mice was much greater than in the weight gain in the other groups (*p* < 0.05).

It is important to prevent body weight gain and fat storage in the treatment of obesity-related complications. We observed that the average body weight of the HFCD + P75 group was significantly lower than that of the HFCD group (Table 1). Nevertheless, both HFCD + PEITC groups consumed more food than the HFCD group, but gained less body weight and body fat. Interestingly, the food intake of mice in the HFCD group was less than that in both HFCD + PEITC groups. Our results demonstrate that supplementation with PEITC significantly prevented body weight gain in HFCD mice.

We observed HFCD-induced fatty liver in C57BL/6 mice. The HFCD group displayed significantly increased liver weight compared with the CON group (*p* < 0.05). PEITC inhibited the increase in liver, epididymal, and retroperitoneal adipose tissues. These results suggest that supplementation with PEITC decreases the severity of HFCD-induced fatty liver.

To explore the inhibitory effect of PEITC on hepatic lipid accumulation, H&E staining was used to observe the histology of liver tissue. Consistent with the results of another study involving mice fed HFD [43], a major histopathological finding of hepatic steatosis was observed in the HFCD group. The H&E histological analysis revealed the presence of large lipid droplets in the liver of HFCD group compared with that of the CON group. PEITC supplementation histological analysis revealed the presence of small lipid droplets in the liver. To analyze the effect of PEITC on atherosclerotic plaque formation, H&E staining was used to observe the histology of aorta tissue. The lumens of blood vessels were narrower in the HFCD group. In contrast, in mice consuming the PEITC supplemented diet, the lumens of blood vessels were similar in size to those in mice in the CON group. Previous studies have shown that significant narrowing of the lumen of blood vessels can cause atherosclerosis [31,44]. Thus, PEITC supplementation may alleviate atherosclerotic plaque areas.

We observed that PEITC supplementation reduced serum AST and ALT levels. Elevation of these serum liver enzymes has been associated with liver injury [45]. In addition, increases in TC, TG, LDL-C, and HOMA-IR are associated with the development of metabolic diseases. All were elevated in the HFCD group and PEITC supplementation counteracted the effects of the HFCD. A direct correlation between LDL and atherosclerosis has been reported [46]. Furthermore, elevated serum levels of TC and TG are also considered important risk factors for the initiation and promotion of atherosclerosis [47]. Therefore, increased cholesterol levels can be a major predictor of CVD development. Other authors described the dramatic increase in TC and LDL-C levels, with only a relatively slight change in HDL-C in animal models fed HFD [48]. Consistent with these observations, PEITC supplementation resulted in decreased serum levels of TC, TG, and LDL-C, and slight changes in HDL-C. In both HFCD + PEITC groups, metabolic indices, including HOMA-IR and AI, were improved compared with those in the HFCD group. Our data indicate that PEITC supplement is effective in improving serum lipid profiles in mice fed HFCD.

Next, to understand the molecular mechanisms whereby PEITC alleviates obesity and atherosclerosis, we performed immunoblotting and qPCR analysis in liver tissue. The reverse cholesterol transport pathway is crucial in reducing cholesterol levels in peripheral tissues and blood as well as inhibiting the formation of atherosclerotic plaques in the aorta [8]. As a key protein regulating the reverse cholesterol transport pathway, ABCA1 modulates lipid metabolism by mediating the reverse transportation of cholesterol to apoA1 and promoting HDL formation [8]. In this context, the critical role of ABCA1 may have a significant impact on the progression of atheromatous plaques [9]. LXRα induces ABCA1 gene expression [8] and PPAR induces transcription of LXRα and ABCA1 to enhance cholesterol efflux [9]. The PPARγ–LXRα–ABCA1 pathway is important in cholesterol homeostasis, lipid metabolism, and inflammation. Thus, the pathway is important in the pathogenesis of atherosclerosis and has promising potential for therapeutic manipulation [49]. Consistent with these prior observations, the present data demonstrate that activation of the reverse cholesterol transport pathway protects mice with HFCD-induced obesity. PEITC supplementation resulted in upregulation of PPARγ, LXR-α, and ABCA1. These findings demonstrate that PEITC can modulate atherosclerotic-related lipid metabolism via the reverse cholesterol transport pathway. Another interesting finding was that PPARγ expression increased in the HFCD group and HFCD + P75 group compared with the CON group (Figure 3). Consistent with our results, a prior study reported that PPARγ stimulates reverse cholesterol transport, allowing hepatic elimination of excessive cellular cholesterol [8]. We believe that the activity of PPARγ is probably modified by environmental factors, such as HFCD, and the upregulation of PPARγ stimulates reverse cholesterol transport to prevent atherosclerosis. PPARγ is normally expressed in both human and murine livers at only 10 to 30% of the levels found in adipose tissue [50]. Further research will be needed to determine the mechanism whereby PPARγ is activated in the liver of obese mice.

To date, in vivo results have revealed that HFD causes lipid accumulation in the liver [51]. CD36 contributes to the process of atherosclerosis because CD36 deficiency prevents plaque development in apoE-/- mice [5]. SR-A1 and CD36 play crucial roles in lipid accumulation and the initiation and development of atherosclerosis [8,52]. In addition, inactivation of the scavenger receptors reduces liver inflammation associated with nonalcoholic fatty hepatitis [53]. In this study, immunoblotting was used to measure the expression of the SR-A1 and CD36 genes, which are associated with lipid accumulation. The expression of both genes was increased in the liver tissues of mice from the HFCD group. Compared with the HFCD group, expression levels were downregulated in mice fed a diet supplemented with PEITC. These findings indicate that PEITC can lessen the development of atherosclerosis and hyperlipidemia by inhibiting lipid accumulation.

Inflammation has been linked mechanistically to obesity and atherosclerosis [54]. It is not surprising that NF-κB influences numerous CVDs, including atherosclerosis. In this study, the expression levels of the NF-κB gene were increased in the HFCD group. Increased NF-κB expression was downregulated by PEITC supplementation. Thus, PEITC supplementation suppressed the inflammatory response by downregulating NF-κB expression.

PETIC may have a protective effect on obesity and atherosclerosis by regulating the expression of PPARγ, LXRα, ABCA1, SR-A1, CD36, and NF-κB. We suggest that PEITC supplementation can effectively alleviate lipid metabolism and reduce inflammatory responses in HFCD obese mice.

We observed that the HFCD enhanced the acetylation status of H3K9 and H3K27 in the liver. These effects were reduced in mice consuming the PEITC supplemented diet. Luo et al. [55] reported that HFD enhanced the acetylation status of H3K9 in adipose tissue. Cordero-Herrera [12] reported that pretreatment with the main cocoa flavanol (−)-epicatechin counteracted the increased acetylation of H3K9 and H3K4 dimethylation and attenuated the diminished H3K9 dimethylation triggered by high glucose concentrations. In this study, H3K9 acetylation and H3K4 dimethylation were reduced with PEITC supplementation. Our data indicate that obesity increases H3K9 acetylation, thereby enhancing monocyte activation. This allows the binding of NF-κB to inflammatory genes. The effects are lessened by PEITC. Another interesting finding was the enhanced status of H3K4 dimethylation and H3K27 di-/trimethylation in the liver of mice consuming the HFCD. These effects were reduced by PEITC dietary supplementation. H3K4 methylation has been correlated with stage-specific progression of atherosclerosis [56]. A global increase in H3K27 trimethylation was also observed in atherosclerotic plaques in later stages of the pathology [39]. In this study, H3K4 methylation and H3K27 trimethylation were reduced with PEITC supplementation. These results demonstrate that PEITC has a beneficial effect on atherosclerotic plaque development that involves the downregulation of H3K9 and H3K27 acetylation, H3K4 dimethylation, and H3K27 di-/trimethylation.

Finally, immunoblotting analysis demonstrated that HDAC3 expression in liver tissue was increased in the HFCD group compared with that in the CON group, whereas PEITC supplementation group showed a decrease in expression similar to the CON group. Hoeksema et al. reported that the deletion of HDAC3 improves atherosclerotic plaque stabilization and inflammation [57]. The collective results suggest that PEITC supplementation can modulate atherosclerotic plaques by inhibiting the expression level of HDAC3 in mice fed HFCD.

## 5. Conclusions

To summarize, we observed that PEITC supplementation improved body weight gain, fatty liver, and evaluated serum levels of lipid associated molecules in C57BL/6 mice fed the HFCD. Liver lipid accumulation and atherosclerotic plaque formation in the aorta were alleviated with PEITC supplementation. All these factors can contribute to the prevention of atherosclerosis and obesity. In addition, PEITC significantly stimulated the reverse cholesterol transport pathway and reduced lipid accumulation and the inflammatory response by modulating PPARγ, LXR-α, ABCA1, SR-A1, CD36, and NF-κB. Moreover, PEITC had beneficial effects on obesity and atherosclerosis via histone modification. We determined the PEITC and GA concentration range based on the evidence provided by several studies [27,41,58,59,60,61]. No adverse events were observed throughout the study. Therefore, we suggest that supplementation with 30 and 75 mg/kg/day PEITC could be beneficial for obesity, related metabolic disorders, and atherosclerosis in C57BL/6 mice. Further studies will be required to prove that PEITC is an efficient anti-obesity and anti-atherosclerosis agent.

## Figures and Tables

**Figure 1 nutrients-12-03657-f001:**
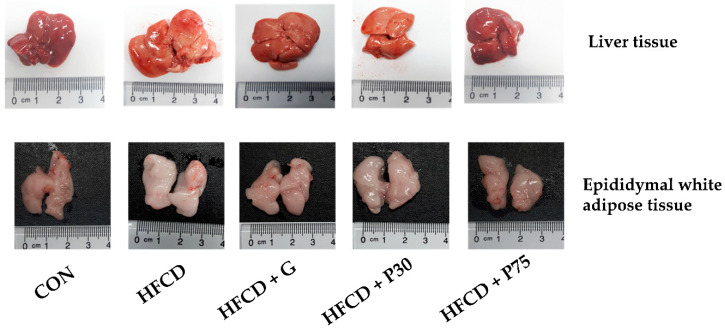
Representative images of liver and epididymal white adipose tissue. Representative macroscopic pictures of male C57BL/6 mice from different groups at the end of the experiment. After sacrifice of the mice, liver and epididymal white adipose tissue were obtained and photographed. Control (CON): AIN-93G diet, high fat/cholesterol diet (HFCD): 60% calories from fat + 1% cholesterol, HFCD + G: HFCD + 3 mg/kg/day gallic acid (GA), HFCD + P30: HFCD + 30 mg/kg/day phenethyl isothiocyanate (PEITC), HFCD + P75: HFCD + 75 mg/kg/day PEITC.

**Figure 2 nutrients-12-03657-f002:**
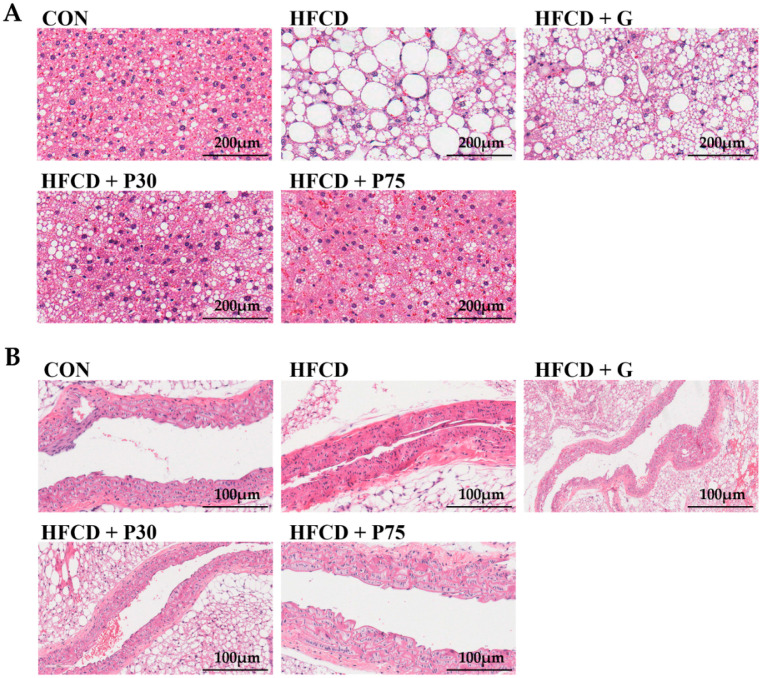
Effect of PEITC supplementation on the livers and aortas of mice: (**A**) Representative photomicrographs of hematoxylin and eosin (H&E) staining of liver tissue. The white areas in the hepatocytes are intracellular lipid droplets (magnification 200× *g*) in H&E staining of formalin-fixed, paraffin-embedded sections; (**B**) H&E staining of aorta tissue (magnification 400× *g*) in sections of formalin-fixed, paraffin-embedded tissue. CON: AIN-93G diet, HFCD: 60% calories from fat + 1% cholesterol, HFCD + G: HFCD + 3 mg/kg/day of GA, HFCD + P30: HFCD + 30 mg/kg/day PEITC, HFCD + P75: HFCD + 75 mg/kg/day PEITC.

**Figure 3 nutrients-12-03657-f003:**
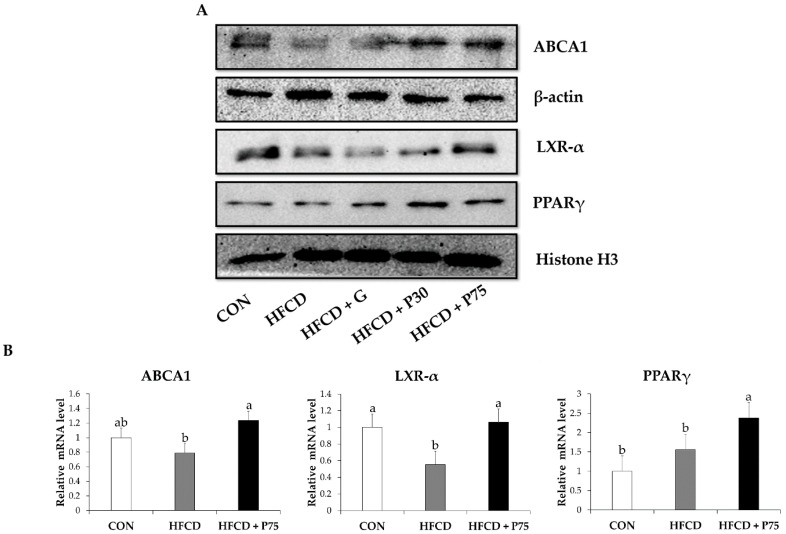
PEITC regulates the reverse cholesterol transport pathway in the livers of HFCD-fed mice: (**A**) Immunoblotting was used to measure the protein levels of ATP binding cassette subfamily A member 1(ABCA1), liver-X-receptor α (LXR-α), and peroxisome proliferator activated receptor gamma (PPARγ). Equal loading of protein was ensured by stripping the immunoblot and reprobing for β-actin and histone H3; (**B**) Evaluated mRNA levels of ABCA1, LXR-α, and PPARγ in mice liver. Values were analyzed by the 2^−ΔΔCT^ method. Significance was determined by comparison with β-actin normalized 2^−ΔΔCT^ values. Data represent the means ± standard deviation (SD, *n* = 3, degrees of freedom = 2), and different letters indicate significant differences (*p* < 0.05, a > b), as determined using Duncan’s multiple range test. CON: AIN-93G diet, HFCD: 60% calories from fat + 1% cholesterol, HFCD + G: HFCD + 3 mg/kg/day GA, HFCD + P30: HFCD + 30 mg/kg/day PEITC, HFCD + P75: HFCD + 75 mg/kg/day PEITC.

**Figure 4 nutrients-12-03657-f004:**
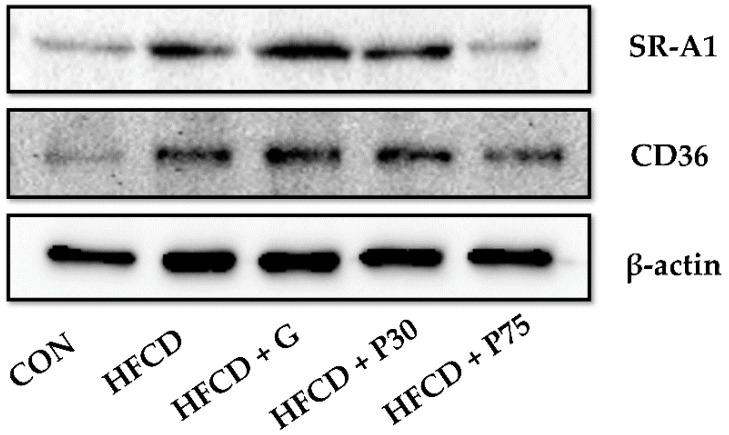
PEITC regulates lipid accumulation genes in the livers of HFCD-fed mice: Immunoblotting was used to measure the protein levels of scavenger receptor A (SR-A1) and cluster of differentiation 36 (CD36). Equal loading of protein was ensured by stripping the immunoblot and reprobing for β-actin. CON: AIN-93G diet, HFCD: 60% calories from fat + 1% cholesterol, HFCD + G: HFCD + 3 mg/kg/day GA, HFCD + P30: HFCD + 30 mg/kg/day PEITC, HFCD + P75: HFCD + 75 mg/kg/day PEITC.

**Figure 5 nutrients-12-03657-f005:**
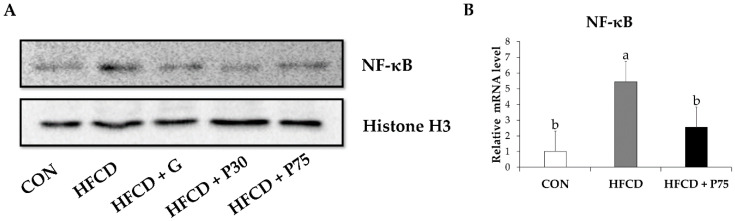
PEITC regulates inflammation-related genes in the livers of HFCD-fed mice: (**A**) Protein levels were measured by immunoblotting for nuclear factor-kappa B (NF-κB). Equal loading of protein was ensured by stripping the immunoblot and reprobing for histone H3; (**B**) Evaluated mRNA levels of NF-κB in mice liver. Values were analyzed by the 2^−ΔΔCT^ method. Significance was determined by comparison with β-actin normalized 2^−ΔΔCT^ values. Data represent the means ± standard deviation (SD, *n* = 3, degrees of freedom = 2), and different letters indicate significant differences (*p* < 0.05, a > b), as determined using Duncan’s multiple range test. CON: AIN-93G diet, HFCD: 60% calories from fat + 1% cholesterol, HFCD + G: HFCD + 3 mg/kg/day GA, HFCD + P30: HFCD + 30 mg/kg/day PEITC, HFCD + P75: HFCD + 75 mg/kg/day PEITC.

**Figure 6 nutrients-12-03657-f006:**
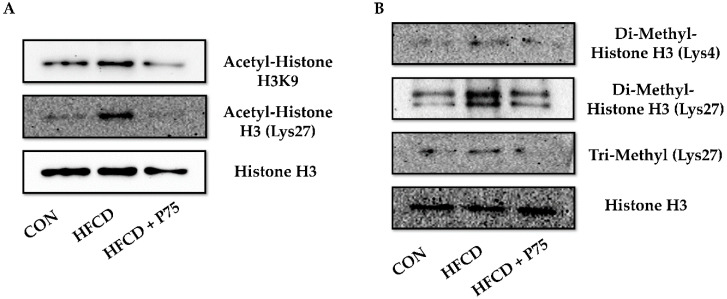
PEITC regulates histone modification in the livers of HFCD-fed mice: (**A**) Immunoblotting was used to measure the levels of H3K9 and H3K27 acetylation; (**B**) Immunoblotting was used to measure the levels of H3K4 dimethylation and H3K27 di-/trimethylation, and equal loading of protein was ensured by stripping the immunoblot and reprobing it for histone H3. CON: AIN-93G diet, HFCD: 60% calories from fat + 1% cholesterol, HFCD + G: HFCD + 3 mg/kg/day GA, HFCD + P30: HFCD + 30 mg/kg/day PEITC, HFCD + P75: HFCD + 75 mg/kg/day PEITC.

**Figure 7 nutrients-12-03657-f007:**
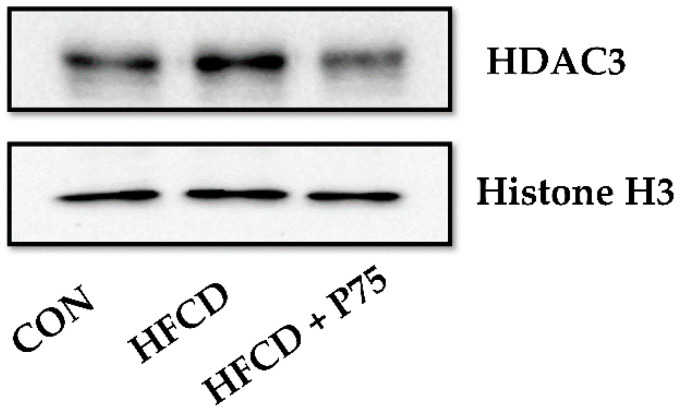
PEITC regulates HDAC3 in the livers of HFCD-fed mice: Immunoblotting was used to measure the levels of HDAC3. Equal loading of protein was ensured by stripping the immunoblot and reprobing it for histone H3. CON: AIN-93G diet, HFCD: 60% calories from fat + 1% cholesterol, HFCD + G: HFCD + 3 mg/kg/day GA, HFCD + P30: HFCD + 30 mg/kg/day PEITC, HFCD + P75: HFCD + 75 mg/kg/day PEITC.

**Table 1 nutrients-12-03657-t001:** Effects of phenethyl isothiocyanate (PEITC) supplementation on the body weight and food efficiency.

	CON	HFCD	HFCD + G	HFCD + P30	HFCD + P75
**Baseline body weight (g)**	19.85 ± 0.33 ^a^	19.87 ± 0.31 ^a^	19.88 ± 0.3 ^a^	19.95 ± 0.33 ^a^	19.87 ± 0.31 ^a^
**Final body weight (g)**	38.03 ± 0.61 ^b^	47.72 ± 0.7 ^a^	44.25 ± 1.89 ^a^	44.68 ± 0.87 ^a^	37.07 ± 1.66 ^b^
**Body weight change (g)**	18.18 ± 0.47 ^b^	27.85 ± 0.55 ^a^	25.03 ± 1.78 ^a^	24.73 ± 0.78 ^b^	17.2 ± 1.55 ^b^
**Food efficiency ratio (%) ^1)^**	6.79 ± 0.17 ^b^	9.55 ± 0.19 ^a^	7.38 ± 0.53 ^b^	5.6 ± 0.18 ^c^	4.63 ± 0.42 ^c^
**Average daily food intake (g/day)**	2.91 ± 0 ^b^	3.17 ± 0.01 ^b^	3.35 ± 0.01 ^ab^	4.36 ± 0.01 ^a^	4.04 ± 0.01 ^ab^

^1)^ Food efficiency ratio (body weight gain (g)/total food intake (g)) × 100. Body weight was measured once every 3 days during the experimental period. Data are expressed as mean ± standard deviation (SD, *n* = 10, degrees of freedom = 4), and different letters indicate significant differences (*p* < 0.05, a > b > c), as determined using Duncan’s multiple range test. Control (CON): AIN-93G diet, high fat/cholesterol diet (HFCD): 60% calories from fat + 1% cholesterol, HFCD + G: HFCD + 3 mg/kg/day gallic acid (GA), HFCD + P30: HFCD + 30 mg/kg/day PEITC, HFCD + P75: HFCD + 75 mg/kg/day PEITC.

**Table 2 nutrients-12-03657-t002:** Effects of PEITC supplementation on liver, adipose, and kidney weight in HFCD.

	CON	HFCD	HFCD + G	HFCD + P30	HFCD + P75
**Liver (g)**	1.45 ± 0.08 ^b^	3.09 ± 0.21 ^a^	2.8 ± 0.16 ^a^	2.7 ± 0.15 ^a^	2.1 ± 0.26 ^b^
**Epididymal white adipose tissue (g)**	1.94 ± 0.07 ^b^	2.1 ± 0.11 ^b^	2.17 ± 0.1 ^b^	2.47 ± 0.09 ^a^	1.81 ± 0.15 ^b^
**Retroperitoneal adipose tissue (g)**	0.76 ± 0.08 ^b^	1.04 ± 0.1 ^a^	1.02 ± 0.08 ^a^	0.98 ± 0.09 ^a^	0.62 ± 0.07 ^b^
**Kidney (g)**	0.35 ± 0.03 ^b^	0.36 ± 0.01 ^a b^	0.4 ± 0.02 ^a^	0.33 ± 0.01 ^b^	0.35 ± 0.02 ^a b^

Organs weights after 13 weeks in C57BL/6 mices. Data are expressed as mean ± standard deviation (SD, *n* = 10, degrees of freedom = 4), and different letters indicate significant differences (*p* < 0.05, a > b), as determined using Duncan’s multiple range test. CON: AIN-93G diet, HFCD: 60% calories from fat + 1% cholesterol, HFCD + G: HFCD + 3 mg/kg/day GA, HFCD + P30: HFCD + 30 mg/kg/day PEITC, HFCD + P75: HFCD + 75 mg/kg/day PEITC.

**Table 3 nutrients-12-03657-t003:** Effects of PEITC supplementation on the serum levels of lipid associated molecules.

	CON	HFCD	HFCD + G	HFCD + P30	HFCD + P75
**AST**	36.48 ± 8.17 ^ab^	50.96 ± 8.87 ^a^	18.53 ± 4.81 ^b^	25.88 ± 6.02 ^b^	44.22 ± 10.32 ^a^
**ALT**	62.05 ± 14.13 ^a^	112.6 ± 19.98 ^a^	85.49 ± 17.57 ^a^	64.76 ± 14.77 ^a^	73.92 ± 16.97 ^a^
**TC (mg/dL)**	69.18 ± 1.96 ^bc^	113.13 ± 4.52 ^a^	76.33 ± 2.51 ^b^	71.05 ± 2.92 ^bc^	67.06 ± 5.75 ^c^
**TG (mg/dL)**	67.39 ± 6.94 ^b^	94.31 ± 7.05 ^a^	76.43 ± 6.59 ^a b^	68.03 ± 5.39 ^b^	68.5 ± 8.45 ^b^
**HDL-C (mg/dL)**	47.38 ± 5.17 ^a^	43.91 ± 4.04 ^a^	41.49 ± 6.18 ^a^	46.32 ± 5.62 ^a^	47.56 ± 2.64 ^a^
**LDL-C (mg/dL)**	23.53 ± 4.9 ^b^	46.97 ± 6.69 ^a^	34.41 ± 6.54 ^ab^	22.31 ± 6.37 ^b^	27.12 ± 6.19 ^ab^
**NEFA**	0.035 ± 0.00 ^a^	0.045 ± 0.01 ^a^	0.032 ± 0.00 ^a^	0.036 ± 0.00 ^a^	0.030 ± 0.00 ^a^
**AI ^1)^**	0.17 ± 0.03 ^b^	0.33 ± 0.03 ^a^	0.33 ± 0.07 ^a^	0.24 ± 0.07 ^ab^	0.14 ± 0.04 ^b^

^1)^ Atherogenic index (log10 (TG/HDL-C)). Data are expressed as mean ± standard deviation (SD, *n* = 10, degrees of freedom = 4), and different letters indicate significant differences (*p* < 0.05, a > b > c), as determined using Duncan’s multiple range test. CON: AIN-93G diet, HFCD: 60% calories from fat + 1% cholesterol, HFCD + G: HFCD + 3 mg/kg/day GA, HFCD + P30: HFCD + 30 mg/kg/day PEITC, HFCD + P75: HFCD + 75 mg/kg/day PEITC. AST, aspartate aminotransferase; ALT, alanine aminotransferase; TC, total cholesterol; TG, triglycerides; HDL-C, high-density lipoprotein cholesterol; LDL-C, low-density lipoprotein cholesterol; NEFA, non-esterified fatty acid.

**Table 4 nutrients-12-03657-t004:** Effects of PEITC supplementation on insulin, blood glucose, and HOMA-IR.

	CON	HFCD	HFCD + G	HFCD + P30	HFCD + P75
**Serum insulin (ng/mL)**	0.47 ± 0.11 ^bc^	1.45 ± 0.15 ^a^	0.83 ± 0.13 ^b^	0.62 ± 0.04 ^bc^	0.31 ± 0.19 ^c^
**Fasting glucose (mg/dL)**	146.67 ± 19.22 ^b^	216 ± 15.04 ^a^	221.67 ± 3.48 ^a^	170 ± 13.65 ^a b^	147.67 ± 33.21 ^b^
**HOMA-IR ^1)^**	0.17 ± 0.06 ^c^	0.77 ± 0.06 ^a^	0.45 ± 0.08 ^b^	0.26 ± 0.02 ^bc^	0.14 ± 0.1 ^c^

^1)^ Homeostatic model assessment of insulin resistance (HOMA-IR) (fasting insulin (μU/mL) × fasting glucose (mmol/L)/22.5). Data are expressed as mean ± standard deviation (SD, *n* = 3, degrees of freedom = 4), and different letters indicate significant differences (*p* < 0.05, a > b > c), as determined using Duncan’s multiple range test. CON: AIN-93G diet, HFCD: 60% calories from fat + 1% cholesterol, HFCD + G: HFCD + 3 mg/kg/day GA, HFCD + P30: HFCD + 30 mg/kg/day PEITC, HFCD + P75: HFCD + 75 mg/kg/day PEITC.

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
