# Peer review of "Phenethyl Isothiocyanate Protects against High Fat/Cholesterol Diet-Induced Obesity and Atherosclerosis in C57BL/6 Mice"

_nutrients, 2020, doi:10.3390/nu12123657_

Round 1
Reviewer 1 Report
The study by Gwon and collaborators showed that the supplementation on a high-fat diet (HFD) with phenethyl isothiocyanate (PEITC) in mice diminished the fat/cholesterol accumulation in liver and aorta and decreased liver HFD-related inflammation. In light of this, authors suggested PEITC could be used as a dietary implementation to prevent atherosclerotic plaques accumulation and a protective effect on obesity.
Several issues listed below need to be addressed by the authors.
- In the introduction, the aim of the study should provide more details. Authors focused their attention on the role of gallic acid (GA) and on the reason why they choose it as positive control. I recommend more details be provided to explain the design and the aim of the study.
- The authors decided to use male instead of female mice. Please incorporate a justification for studying only males given that nonalcoholic fatty liver disease, obesity and atherosclerosis are also very common in females (see: Man et al., 2020; Mauvais-Jarvis et al., 2015; Leonardo et al., 2019).
- Did the authors collect plasma or serum samples? In the manuscript, it is not clear whether serum or plasma are being used. For example, section 2.2. title refers to plasma (Collection of plasma and tissue samples), while the paragraph refers to serum. Samples of plasma and serum undergo different collection procedures. Please provide an explanation on why two anticoagulants were used and specify when serum or plasma were collected and for which analysis.
- Glucose levels were measured in fasting animals. Could the authors explain why 12 hours fasting were necessary for the measurements reported here? For glucose or insulin tests a fasting duration of 5-6 h has been recommended and is comparable to the overnight fasting in human. In mice an overnight fasting is not considered a basal steady-state as a consequence of the nightly eating habits (see: Schork et al., 2012 or Jensen et al., 2013).
- Why the authors choose the glucose meter and not a biochemical assay to measure glucose blood levels? Which glucometer did they use? The use of glucometer is long debated for the less accuracy (see: Togashi et al., 2016, Morley et al., 2018). The authors should provide a rationale in the text regarding this issue.
- Statistical analysis: when reporting the results, the degree of freedom must be reported.
- Based on the results, authors compared mean values to others mean values and not only to the corresponding control group mean. In light of this, Dunnett post-hoc test seemed to be not appropriate for this purpose. The authors should provide a rationale regarding their choice.
- Line 22. The sentence “ Hepatic lipid … H&E staining” is inconsistent and contrary to the result they presented. Please clarify.
- Line 44. Please add “is” between “expression” and “significantly”.
- Line 45. The sentence “Atherosclerosis is a chronic disease in which lipid particles infiltrate into the arterial wall” sounds like a repetition of the sentence in lines 35-36 “…characterized by the infiltration of lipid particles into the arterial wall”.
- Line 58. Please correct “It” with “Its”.
- Line 59. Please add “s” to “plaque”.
- Line 111. How did the authors collect the blood?
- Line 143 and 150. Please specify the centrifuge parameters (speed, time, temperature).
- Line 240. Please avoid starting a sentence with “But”.
- Line 282. Please correct “on the level of the serum levels”.
- Line 416. The sentence “Interestingly… HFD+PEITC” should be rephrased in a more clear way.
Author Response
"Please see the attachment also"
To
The Editor
NUTRINES
Dear Dr. Sasa Cakara,
We appreciate the careful and thorough review of our manuscript entitled “Phenethyl Isothiocyanate Protects Against High Fat/Cholesterol Diet-Induced Obesity and Atherosclerosis in C57BL/6 Mice” by Min-Hee Gwon, Young-Sun Im, A-Reum Seo, Kyoung Yun Kim, Ha-Rin Moon and Jung‐Mi Yun submitted for publication in Nutrients. We are pleased to note that this manuscript maybe considered for publication in Nutrients after improvements and clarifications. As suggested, we have revised the manuscript based on suggestions provided by reviewers.
We trust that the revised manuscript will now be acceptable for publication in Nutrients.
I look forward to hearing from you at your earliest convenience.
Sincerely,
Jung-Mi Yun, PhD
Professor, Department of Food and Nutrition, Chonnam National University
Yongbong-ro, Buk-gu, Gwangju 61186, South Korea
Tel: +82-62-530-1332
Fax: +82-62-530-1339
E-mail: [email protected]
First of all, we thank the editor and reviewers for the careful review of our manuscript.
As point out by reviewers, we revised the manuscript and we hope this now meets the satisfaction of reviewers. We used the "Track Changes" function in Microsoft Word in revised manuscript for corrections.
Responses to reviewer #1:
First of all, we are grateful to the reviewer for the careful review of our manuscript
As point out by the reviewer, we revised the manuscript.
Comment #1: In the introduction, the aim of the study should provide more details. Authors focused their attention on the role of gallic acid (GA) and on the reason why they choose it as positive control. I recommend more details be provided to explain the design and the aim of the study.
Response: Per the reviewer’s suggestion, we revised and presented the purpose of the research in detail on page 3, line 98–line 104 in the revised manuscript, as shown below:
“In this study, we hypothesized that HFCD supplementation with PEITC in mice could reduce fat/cholesterol in the liver and aorta and has a protective effect against obesity and atherosclerosis. To test our hypothesis, we used C57BL/6 mice with high fat/cholesterol-induced obesity and analyzed the effect of PEITC supplementation through the expression of genes related to inflammation, reverse cholesterol transport, and lipid accumulation pathways in the liver. Furthermore, we analyzed H3K9 and H3K27 acetylation, H3K4 dimethylation, and H3K27 di-/trimethylation with modulated PEITC supplementation.”
Comment #2: The authors decided to use male instead of female mice. Please incorporate a justification for studying only males given that nonalcoholic fatty liver disease, obesity and atherosclerosis are also very common in females (see: Man et al., 2020; Mauvais-Jarvis et al., 2015; Leonardo et al., 2019).
Response: We thank you for this comment. As mentioned, obesity and cardiovascular disease occur differently in the two sexes because of various biological factors differing between the sexes. We decided to use male mice, referring to the study by Richard Kirsch et al. [Kirsch et al 2003]. They used both male and female C57/BL6 mice in their experiments and reported that male C57/BL6 mice developed the histological features that most closely resemble those seen in human NASH. Additionally, they reported that male C57/BL6 mice developed a higher degree of inflammation, necrosis, and lipid peroxidation than the female mice. NASH tends to develop in people who are overweight or obese, or have diabetes, high cholesterol or high triglycerides. NASH is also associated with endothelial dysfunction, which is a marker of early atherosclerosis [Villanova et al 2005 & Schindhelm et al 2005 & Senturk et al 2008]. The vascular abnormalities observed in NASH patients, are similar to those seen in patients with CVD, impairing the vascular response to physiologic and pharmacologic stimuli [Bieghs et al 2012].
Ref:
Kirsch, R.; Clarkson, V.; Shephard, E.G.; Marais, D.A.; Jaffer, M.A.; Woodburne, V.E.; Kirsch, R.E.; Hall Pde, L. Rodent nutritional model of non-alcoholic steatohepatitis: species, strain and sex difference studies. J Gastroenterol Hepatol 2003, 18, 1272-1282, doi:10.1046/j.1440-1746.2003.03198.x.
Villanova, N.; Moscatiello, S.; Ramilli, S.; Bugianesi, E.; Magalotti, D.; Vanni, E.; Zoli, M.; Marchesini, G. Endothelial dysfunction and cardiovascular risk profile in nonalcoholic fatty liver disease. Hepatology 2005, 42, 473-480, doi:10.1002/hep.20781.
Schindhelm, R.K.; Diamant, M.; Bakker, S.J.; van Dijk, R.A.; Scheffer, P.G.; Teerlink, T.; Kostense, P.J.; Heine, R.J. Liver alanine aminotransferase, insulin resistance and endothelial dysfunction in normotriglyceridaemic subjects with type 2 diabetes mellitus. Eur J Clin Invest 2005, 35, 369-374, doi:10.1111/j.1365-2362.2005.01502.x.
Senturk, O.; Kocaman, O.; Hulagu, S.; Sahin, T.; Aygun, C.; Konduk, T.; Celebi, A. endothelial dysfunction in turkish patients with non-alcoholic fatty liver disease. Intern Med J 2008, 38, 183-189, doi:10.1111/j.1445-5994.2007.01481.x
Bieghs, V.; Rensen, P.C.; Hofker, M.H.; Shiri-Sverdlov, R. NASH and atherosclerosis are two aspects of a shared disease: central role for macrophages. Atherosclerosis 2012, 220, 287-293, doi:10.1016/j.atherosclerosis.2011.08.041.
Comment 3#: Did the authors collect plasma or serum samples? In the manuscript, it is not clear whether serum or plasma are being used. For example, section 2.2. title refers to plasma (Collection of plasma and tissue samples), while the paragraph refers to serum. Samples of plasma and serum undergo different collection procedures. Please provide an explanation on why two anticoagulants were used and specify when serum or plasma were collected and for which analysis.
Response: We used serum in our experiments. Thus, we consistently revised the term to “serum” in the modified manuscript, including the section 2.2 heading. Additionally, the anticoagulants were removed from the revised manuscript.
Comment 4#: Glucose levels were measured in fasting animals. Could the authors explain why 12 hours fasting were necessary for the measurements reported here? For glucose or insulin tests a fasting duration of 5-6 h has been recommended and is comparable to the overnight fasting in human. In mice an overnight fasting is not considered a basal steady-state as a consequence of the nightly eating habits (see: Schork et al., 2012 or Jensen et al., 2013).
Response: We measured glucose levels after a 12-h fast and not overnight, as studies have used the 12-h fast method [Lee et al 2020 & List et al 2009]. As suggested, there is no issue with this method because studies still use 12-h fasting method; nevertheless, we will keep in mind the concern of animal welfare in the future.
Ref:
Lee, H.A.; Cho, J.H.; Afinanisa, Q.; An, G.H.; Han, J.G.; Kang, H.J.; Choi, S.H.; Seong, H.A. Ganoderma lucidum extract reduces insulin resistance by enhancing AMPK activation in high-fat diet-induced obese mice. Nutrients 2020, 12, doi:10.3390/nu12113338.
List, E.O.; Palmer, A.J.; Berryman, D.E.; Bower, B.; Kelder, B.; Kopchick, J.J. Growth hormone improves body composition, fasting blood glucose, glucose tolerance and liver triacylglycerol in a mouse model of diet-induced obesity and type 2 diabetes. Diabetologia 2009, 52, 1647-1655, doi:10.1007/s00125-009-1402-z.
Comment 5#: Why the authors choose the glucose meter and not a biochemical assay to measure glucose blood levels? Which glucometer did they use? The use of glucometer is long debated for the less accuracy (see: Togashi et al., 2016, Morley et al., 2018). The authors should provide a rationale in the text regarding this issue.
Response: Currently, there is controversy regarding the accuracy of the blood glucose meter; however, until now, glucose meters have been widely used in studies conducted using animal models [Shi et al 2019 & Nguyen et al 2019 & Jung et al 2020]. Glucometers are useful for screening drugs; evaluating the efficacy of intervention with diets, exercise, or drugs; and assessing the genetic impact in preclinical studies. Many studies have used blood glucose meters [Togashi et al 2016].
Because glucose is dissolved in a water component in the blood, the whole-blood glucose concentration is lower than that in the plasma by 10%. Accordingly, the difference in the distribution of water could cause the measurement discrepancies between glucometers and the laboratory biochemical method. However, according to YuTogashi et al. [Togashi et al 2016], a difference may occur between blood glucose levels measured by a glucometer and those measured via laboratory biochemical tests in mice; however, above all, the reproducibility and consistency of measurements using a glucometer are more important.
Since we used the same volume of blood and the same glucose meter (CareSens® N (i-SENS, Inc., Seoul, South Korea)) in this study and performed the test several times, we obtained the same quality and consistent measurements using a glucometer. We added the following sentence in the methods section of the revised manuscript on page 3, line 136:
“The portable blood glucose meter CareSens® N (i-SENS, Inc., Seoul, South Korea) was used to measure fasting blood glucose, according to the method described by Shi et al. [revised manuscript #30].”
Ref:
Shi, X.; Zhou, X.; Chu, X.; Wang, J.; Xie, B.; Ge, J.; Guo, Y.; Li, X.; Yang, G. Allicin improves metabolism in high-fat diet-induced obese mice by modulating the gut microbiota. Nutrients 2019, 11, doi:10.3390/nu11122909.
Nguyen, L.V.; Ta, Q.V.; Dang, T.B.; Nguyen, P.H.; Nguyen, T.; Pham, T.V.H.; Nguyen, T.H.; Baker, S.; Le Tran, T.; Yang, D.J., et al. Carvedilol improves glucose tolerance and insulin sensitivity in treatment of adrenergic overdrive in high fat diet-induced obesity in mice. PLoS One 2019, 14, e0224674, doi:10.1371/journal.pone.0224674.
Jung, D.Y.; Kim, J.H.; Jung, M.H. Anti-obesity effects of tanshinone I from salvia miltiorrhiza bunge in mice fed a high-fat diet through inhibition of early adipogenesis. Nutrients 2020, 12, doi:10.3390/nu12051242.
Togashi, Y.; Shirakawa, J.; Okuyama, T.; Yamazaki, S.; Kyohara, M.; Miyazawa, A.; Suzuki, T.; Hamada, M.; Terauchi, Y. Evaluation of the appropriateness of using glucometers for measuring the blood glucose levels in mice. Sci Rep 2016, 6, 25465, doi:10.1038/srep25465.
Comment 6#: Statistical analysis: when reporting the results, the degree of freedom must be reported.
Response: Per the reviewer’s suggestion, we added the degree of freedom to all table and figure legends of the revised manuscript.
Comment 7#: Based on the results, authors compared mean values to others mean values and not only to the corresponding control group mean. In light of this, Dunnett post-hoc test seemed to be not appropriate for this purpose. The authors should provide a rationale regarding their choice.
Response: Per the reviewer’s suggestion, we recalculated using the Duncan's multiple range test. We revised the statistical method on page 5, line 200, and revised the manuscript accordingly, including Tables 1–4, Figure 3, and Figure 5.
Comment 8#: Line 22. The sentence “Hepatic lipid … H&E staining” is inconsistent and contrary to the result they presented. Please clarify.
Response: Per the reviewer’s suggestion, we revised the word “higher” to “lower” in the sentence on page 1, line 21, of the revised manuscript, as below:
“Hepatic lipid accumulation and atherosclerotic plaque formation in the aorta were significantly lower both in HFCD + PEITC than in the HFCD, as revealed by H&E staining”
Comment 9#: Line 44. Please add “is” between “expression” and “significantly”.
Response: Per the reviewer’s suggestion, we added "is" between "expression" and "significantly" on page1, line 43 of the revised manuscript, as below:
“Hepatic CD36 expression is significantly elevated and is considered to be positively correlated with liver fat content in vivo”.
Comment 10#: Line 45. The sentence “Atherosclerosis is a chronic disease in which lipid particles infiltrate into the arterial wall” sounds like a repetition of the sentence in lines 35-36 “…characterized by the infiltration of lipid particles into the arterial wall”.
Response: Per the reviewer’s suggestion, we deleted on page2, line 45 in the revised manuscript.
Comment 11#: Line 58. Please correct “It” with “Its”.
Response: Per the reviewer’s suggestion, we revised "It” to “Its” on page2, line59 in the revised manuscript, as below:
“Its activation-mediated signaling has been detected at different stages of atherosclerosis, beginning with plaque formation, destabilization, and rupture”
Comment 12#: Line 59. Please add “s” to “plaque”.
Response: Per the reviewer’s suggestion, we revised "plaque” to “plaques” in line 59 in the revised manuscript, as below:
“Its activation-mediated signaling has been detected at different stages of atherosclerosis, beginning with plaque formation, destabilization, and rupture”
Comment 13#: Line 111. How did the authors collect the blood?
Response: We collected blood samples from the postcaval vein of each animal immediately after sacrifice;, and this was added on page3, line 122 in the revised manuscript:. “Blood samples were collected from the postcaval vein of each animal immediately after sacrifice and centrifuged at 870 g for 5 min at 4 °C.”
Comment 14#: Line 143 and 150. Please specify the centrifuge parameters (speed, time, temperature).
Response: Per the reviewer’s suggestion, we indicated the speed in terms of “g,” the time in “min” or “h,” and the temperature in “°C” in the centrifuge parameters on lines 153, 160, and 164 in the revised manuscript as below:
Line 153: “The lysates were collected and cleared by centrifugation at 16,343 g for 1 min at 4 °C, and the supernatant was aliquoted and stored at −80 °C.”
Line 160: “The extract was centrifuged at 16,000 g for 5 min at 4 °C, and the pH was adjusted to 8 using a neutralization buffer.”
Line 164: The next day, the samples were centrifuged at 16,000 g for 1 h at 4 °C and washed with acetone and PCA.”
Comment 15#: Line 240. Please avoid starting a sentence with “But”.
Response: Per the reviewer’s suggestion, we deleted “But” on page 7, line 253.
“The HFCD + P75 and the CON groups were very similar.”
Comment 16#: Line 282. Please correct “on the level of the serum levels”.
Response: Per the reviewer’s suggestion, we revised " on the levels of the serum levels” to “on the level of the serum levels” in line 59 in the revised manuscript, as below:
“Table 3. Effects of PEITC supplementation on the serum levels of lipid associated molecules.”
Comment 17#: Line 416. The sentence “Interestingly… HFD+PEITC” should be rephrased in a more clear way.
Response: As suggested, we deleted “probably because of the higher caloric density” on page13, line 429 of the revised manuscript for improved clarity, as below:
“Interestingly, the food intake of mice in the HFCD group was less than that in both the HFCD + PEITC groups.”

Reviewer 2 Report
Gwon et al. report anti-obesity and anti-atherosclerosis effects of phenethyl isothiocyanate with associated underlying mechanisms in mice. The study is interesting, please see my comments below:
Methods:
Line 100 – provide a value or range for the constant temperature
Line 111 – how are the fasting blood sugar levels random if they mice have been fasting for 12 h (line 112)?
Line 112 – what dose/volume of diethyl ether was used? This needs to be include so the study is replicable.
Line 128 – past tense; change “is used” to “was used” twice in line 128
Line 135 – “Paraffin blocks make sectioning to a thickness of approximately”. Change to “Paraffin blocks were sectioned to a thickness of approximately”.
Results:
Scalebars in Figure 2 are missing or difficult to see.
All Table and Figure legends have HFCD: 60% kcal from fat. The unit kcal should be written as calories because it is irrelevant to use the kilo prefix unit when describing a percentage.
Discussion:
Lines 426 and 427 – the differences in lipid deposition in the livers between control and treatments needs to be quantified. A “notable increase” is vague and subjective.
Line 508 – “Hoeksema et al., reported that deletion of 509 HDAC3 improves atherosclerotic plaque stabilization and anti-inflammatory”. Does not make sense. Change anti-inflammatory to inflammation?
Line 512-522 – suggest use “Conclusion” subheading to clearly show the findings of this study and distinguish it from the “Discussion” section.
Line 520 – “we suggest that supplementation with 30 and 75 mg/kg/day PEITC could be beneficial for obesity, related metabolic disorders, and atherosclerosis”. Make explicit that this is a mouse study. Why were these doses used for PEITC and 3 mg/kg/day GA? What is the dose translation to humans? Are these realistically achievable in the diet of humans?
References:
Check for consistent use of capital letters in titles. For example, references 2, 6, 11, 13, 14, 15, 17, 20, 21, 25, 32, 40, 54 use capital letters for each word in the title. This should be in sentence case as in references 1, 3, 4 and others.
Reference 36 “Solidago virgaurea” needs to be in italics (genus species) by convention.
Author Response
"Please see the attachment also"
To
The Editor
NUTRINES
Dear Dr. Sasa Cakara,
We appreciate the careful and thorough review of our manuscript entitled “Phenethyl Isothiocyanate Protects Against High Fat/Cholesterol Diet-Induced Obesity and Atherosclerosis in C57BL/6 Mice” by Min-Hee Gwon, Young-Sun Im, A-Reum Seo, Kyoung Yun Kim, Ha-Rin Moon and Jung‐Mi Yun submitted for publication in Nutrients. We are pleased to note that this manuscript maybe considered for publication in Nutrients after improvements and clarifications. As suggested, we have revised the manuscript based on suggestions provided by reviewers.
We trust that the revised manuscript will now be acceptable for publication in Nutrients.
I look forward to hearing from you at your earliest convenience.
Sincerely,
Jung-Mi Yun, PhD
Professor, Department of Food and Nutrition, Chonnam National University
Yongbong-ro, Buk-gu, Gwangju 61186, South Korea
Tel: +82-62-530-1332
Fax: +82-62-530-1339
E-mail: [email protected]
First of all, we thank the editor and reviewers for the careful review of our manuscript.
As point out by reviewers, we revised the manuscript and we hope this now meets the satisfaction of reviewers. We used the "Track Changes" function in Microsoft Word in revised manuscript for corrections.
Responses to reviewer #2:
First of all, we are grateful to the reviewer for the careful review of our manuscript
As point out by the reviewer, we revised the manuscript.
Comment 1#: Line 100. provide a value or range for the constant temperature
Response: Per the reviewer’s suggestion, we added the temperature and humidity range on page 3, line 108, of the revised manuscript, as below:
“The animals were housed under controlled laboratory conditions (20–25 °C; humidity, 55%–60%), under a 12-h shift of the light/dark cycle, with free access to food and water.”
Comment 2#: Line 111 how are the fasting blood sugar levels random if they mice have been fasting for 12 h (line 112)?
Response: We measured fasting blood glucose using the portable blood glucose meter CareSens® N (i-SENS, Inc., Seoul, South Korea) and per reviewer’s suggestion, have deleted the word “random” in the sentence on page 3, line 120, of the revised manuscript, as below:
“Prior to sacrifice, the total body weight, food intake, and fasting blood glucose levels were measured.”
Comment 3#: Line 112 what dose/volume of diethyl ether was used? This needs to be include so the study is replicable.
Response: Per the reviewer’s suggestion, we added the diethyl ether volume oin page 3, line 122, of the revised manuscript, as the below:
“After fasting overnight for 12 h, each mouse was deeply anesthetized with approximately 5 mL of diethyl ether.”
Comment 4#: Line 128 past tense; change “is used” to “was used” twice in line 128
Response: We changed “is used” to “was used” at both instances on page 3, lines 136–138, of the revised manuscript, as below:
“Insulin enzyme-linked immunosorbent assay kit (Morinaga Institute of Biological Science, Yokohama, Japan) was used to measure serum insulin. The portable blood glucose meter CareSens® N (i-SENS, Inc., Seoul, South Korea) was used to measure fasting blood glucose, according to the method described by Shi et al [revised manuscript #30].”
Comment 5#: Line 135 “Paraffin blocks make sectioning to a thickness of approximately”. Change to “Paraffin blocks were sectioned to a thickness of approximately”.
Response: Per the reviewer’s suggestion, we changed “Paraffin blocks make sectioning to a thickness of approximately” to “Paraffin blocks were sectioned to a thickness of approximately” on page 4, line 144, of the revised manuscript, as below:
“Paraffin blocks were sectioned to a thickness of approximately 4–5 µm, mounted on slides, and stained with hematoxylin and eosin (H&E).”
Comment 6#: Scalebars in Figure 2 are missing or difficult to see.
Response: We added scalebars in Figure 2 on page 7 of the revised manuscript.
Comment 7#: All Table and Figure legends have HFCD: 60% kcal from fat. The unit kcal should be written as calories because it is irrelevant to use the kilo prefix unit when describing a percentage.
Response: Per the reviewer’s suggestion, we corrected “HFCD: 60% kcal” to “HFCD: 60% calories” in the revised manuscript, as below:
“normal diet with AIN-93G diet (control, CON), high fat diet (60% calories from fat) containing 1% cholesterol (HFCD), HFCD with 3 mg/kg/day GA (HFCD + G), HFCD with 30 mg/kg/day PEITC (HFCD + P30), or HFCD with 75 mg/kg/day PEITC (HFCD + P75).”
Comment 8#: Lines 426 and 427 the differences in lipid deposition in the livers between control and treatments needs to be quantified. A “notable increase” is vague and subjective.
Response: To improve clarity, we changed “a notable increase in lipid deposition” to “the presence of large lipid droplets” to improve clarity on page 13, line 439-441, and changed “significantly reduced HFCD-induced lipid deposition” to “histological analysis revealed the presence of small lipid droplets” on page 13, line 438, of the revised manuscript, as below:
“The H&E histological analysis revealed the presence of large lipid droplets in the liver of HFCD group compared to that of the CON group.”
“PEITC supplementation histological analysis revealed the presence of small lipid droplets in the liver.”
Comment 9#: Line 508 “Hoeksema et al., reported that deletion of 509 HDAC3 improves atherosclerotic plaque stabilization and anti-inflammatory”. Does not make sense. Change anti-inflammatory to inflammation?
Response: We changed “anti-inflammatory” to “inflammation” on page 15, line 528, of the revised manuscript, as below:
“Hoeksema et al., reported that the deletion of HDAC3 improves atherosclerotic plaque stabilization and inflammation”
Comment 10#: Line 512-522 suggest use “Conclusion” subheading to clearly show the findings of this study and distinguish it from the “Discussion” section.
Response: Per the reviewer’s suggestion, we divided the text in lines 531 on page 15, in the revised manuscript, into a conclusion section and discussion section.
Comment 11#: Line 520 “we suggest that supplementation with 30 and 75 mg/kg/day PEITC could be beneficial for obesity, related metabolic disorders, and atherosclerosis”. Make explicit that this is a mouse study. Why were these doses used for PEITC and 3 mg/kg/day GA? What is the dose translation to humans? Are these realistically achievable in the diet of humans?
Response:
We determined the PEITC and GA concentration range based on the evidence provided by several studies (revised manuscript #27,41,58-61). When these doses in mice (30 and 75 mg/(kg·day)) were converted using the difference in body surface area between humans and mice, the equivalent doses in humans were 40 and 100 mg per serving, respectively [Reagan-Shaw et al 2008]. We wrote this on page 15, line 539, of the revised manuscript, as below:
“We determined the PEITC and GA concentration range based on the evidence provided by several studies [revised manuscript #27,41,58-61].”
Ref:
Reagan-Shaw, S.; Nihal, M.; Ahmad, N. Dose translation from animal to human studies revisited. Faseb j 2008, 22, 659-661, doi:10.1096/fj.07-9574LSF.
Comment 12#: Check for consistent use of capital letters in titles. For example, references 2, 6, 11, 13, 14, 15, 17, 20, 21, 25, 32, 40, 54 use capital letters for each word in the title. This should be in sentence case as in references 1, 3, 4 and others.
Response: Per the reviewer’s suggestion, we consistently used capital letters for each word in the title, including references 2, 6, 11, 13, 14, 15, 17, 20, 21, 25, 30, 33, 40, 41, 49, 55, and 60 in the revised manuscript.
Comment 13#: Reference 36 “Solidago virgaurea” needs to be in italics (genus species) by convention.
Response: Per the reviewer’s suggestion, we italicized "Solidago virgaurea" [revised manuscript #37]
